# Overdurability and Technical Wear of Materials Used in the Construction of Old Buildings

**DOI:** 10.3390/ma14020378

**Published:** 2021-01-14

**Authors:** Jarosław Konior

**Affiliations:** Department of Building Engineering, Faculty of Civil Engineering, Wroclaw University of Science and Technology, Wybrzeże Stanisława Wyspiańskiego 27, 50-370 Wrocław, Poland; jaroslaw.konior@pwr.edu.pl

**Keywords:** tenement houses, technical wear, maintenance, durability, distribution function of the normal distribution

## Abstract

The technical maintenance of old tenement houses traditionally constructed is an ongoing problem, and will continue to be so in the coming years. The subject of the article includes old residential buildings from the turn of the XIX and XX centuries, which are a part of Wroclaw’s downtown district. They can be understood as an essential link in the process of shaping the cultural and social microenvironment of man. The ability of them to meet the multiple expectations of residents depends on the natural aging of tenement houses’ materials, the methods of their maintenance and use, and the influence of the many factors that cause their accelerated wear. The assumed durability is the main reference parameter of the changing age of the inspected tenement houses. The course of the theoretical and observed degree of the technical wear of these buildings was compared with their durability. For the age of these buildings, the technical wear should reach 100%. It was observed that in the first period of use of tenement houses, the phenomenon of “infradurability” occurs, and after exceeding a certain age—depending on the maintenance conditions of the building—the phenomenon of “overdurability” of the building occurs. It was shown that the durability of important elements of old buildings, as a parameter that was defined “a posteriori”, ranges from 153 to 177 years, and is greater than the corresponding literature values indicated “a priori”. The probability of reaching such an age of an element, in which the observed values of technical wear exceed the theoretical values, is much higher than the probability of an opposite event. A comparative analysis of the distribution functions of these probabilities indicates that the probabilities of theoretical wear values are higher than those observed in the case of the assumed literature durability of elements. There is also an inverse relationship for durability that corresponds to the age of the oldest examined elements of tenement houses.

## 1. Introduction

### 1.1. Literature Review

The durability of a building structure is defined as the ability to maintain the users’ requirements for the allowed timeline under the influence of certain factors. The measurer of the durability is the time during which the object retains its properties. The durability of a building depends on many factors, including input assumptions, e.g., the properties of the built-in materials, the accuracy of the executed design, the quality of the performed construction works, and the conditions occurring during the operation of the building, i.e., the way of using it, and the impact of the external environment [1,2]. The demand of assumed durability of a building structure is met if the structure maintains its serviceability, load-bearing capacity and stability throughout its intended service life, without any significant reduction of its serviceability, and also without excessive, unforeseen maintenance costs. Additionally, the materials should fulfil certain functions within the structure [3].

Durability, as one of the basic parameters of the value function, is a concept widely described in technical literature [4,5,6,7,8,9]. The article presents the essence of the durability of residential buildings with regards to an a priori factor, i.e., an assumed factor (assumed before the fact), for forecasting the amount of wear. In publications devoted to the problems of durability in the construction industry, there are many definitions of durability, which differ in their methodological assumptions [10]. Currently, when defining durability, the intensive correlation between the maintenance of a building object and its features, and the ability to maintain technical efficiency, is emphasized [11]. The technical durability of a building is therefore a comprehensive result of the degrees of resistance of components and constituent materials to the effects of time. The process of losing durability occurs gradually and results from the aging of materials under the influence of operational processes and environmental impacts. Within certain limits, durability is an arbitrary feature, which depends on the actual stringency of the requirements in this field.

In papers [12,13,14,15], the concept of standard and operational durability was introduced. Standard durability is the average durability determined or expected for buildings, for their components, or for the construction products that were designed, constructed and operated in accordance with all technical requirements included in the relevant standards, instructions, certificates, etc. Operational durability is the durability of materials or products that were used as intended and subjected to the effects of the most common and typical destructive factors during the building’s operation. Operational durability can be either guaranteed or estimated. Guaranteed durability is the minimum durability that should be achieved by at least 95% of the entire population of elements, while estimated durability is the actual average durability of the components that were built-in in the entire population (excluding defective products). The term “estimated” means that it is not an average value that is calculated according to the rules of mathematical statistics and based on a sufficiently large sample, but is instead determined based on the observations of the performance of an evaluated element.

The durability of load-bearing elements is therefore determined by two parameters: the ultimate limit state (strength and stability) and the limit of stiffness (serviceability limit state) [16]. Durability is an extensive trait that allows the use of other features, including reliability and efficiency (as intensive features). It can therefore be concluded that the durability of building materials and their components is determined by the length of the period in which a properly operated product remains fit for its intended use. According to the theory of reliability, this suitability expresses the product’s readiness to provide the expected portion of the serviceability value. A decrease in serviceability below the established requirements is tantamount to the element’s being unfit for use. This unfitness (wear) means a partial or complete loss of properties that are decisive for the product’s serviceability, and it indicates the need for renovation. In such a case there is also a partial or complete loss of durability [17,18].

A major problem in adopting the durability periods of buildings and their elements is the failure to publish guidelines and instructions for various construction sectors. The durability periods of various building materials, and the structural elements that are composed of them, should refer to the local market. In Poland, such standards concerning assumed durability were periodically published in the 1980s and 1990s in manuals by *Arendarski* [4,5], *Thierry* and *Zaleski* [6], *Bobociński* [12,13], and *Ściścielewski* [19,20,21]. Due to the fact that the research sample discussed in the article concerns old tenement houses with traditional construction, it seems to be entirely legitimate to assume the durability periods given by these authors.

However, the article takes into account an interesting approach of authors such as *Nowogońska* [8,9], *Saleh* [14,15], *Frangopol* [22], *Silva* [23] and *Plebankiewicz* [24,25], which was published in recent years. It considers the entire life cycle of a building object, even after exceeding its assumed durability, up to the so-called technical death of the building.

### 1.2. Durability within the Maintenance

The required or designed durability of a residential building, even if it was designed and built while taking into account the requirements of durability, can only be achieved if during its use it is subjected to such preservation procedures that result from its appropriate maintenance [22]. Therefore, one of the conditions for meeting the performance requirements is an appropriate exploitation program, which should be an element of the project documentation [19,21].

The maintenance of a structure is essential for the owner of the facility. If during its use, maintenance procedures are abandoned, or not performed in a timely manner, the value of the building will decrease [9,23]. However, the essential, from the point of view of durability, is the fact that expenditure on its maintenance will increase [24,25,26]. Unfortunately, investors are usually unaware of the results of the incorrect maintenance, or even of its complete abandonment.

The exploitation program of a building object, which should be a result of arrangements between the designer and the investor, should take into account the following issues:the determination of the functions to be fulfilled by individual parts of the building, and also the permissible changes to these requirements during its use;the specification of the environmental conditions and reasons that will affect parts of the building and its elements;the adoption of the maintenance level for the building and its elements;the determination of the entire service life of the building, or its parts, and also the inter-renovation cycles in which its elements will be replaced.

In addition, the exploitation program of a building object should have a plan of all the treatments concerning the elements that must be maintained or repaired. Some of the repairs result from the conditions of using residential buildings, and their scope may be determined on the basis of the inspection of facilities. The obligation of conducting inspections results from the necessity of appropriate maintenance of building structures, which was defined in Polish Construction Law [27]. According to this law, “an operator of a building object is supposed to use the object in accordance with its intended use and environmental protection demands, and also needs to maintain it in a proper technical and aesthetic condition”.

This document also specifies certain requirements. Those related to the investigation of the technical condition of the components and installations of a residential building that are exposed to harmful atmospheric influences and damaging factors appearing at the use of the building can be considered to be important with regards to durability. Due to the fact that all the impacts on a building should be thought about when considering the degree of the durability of a residential building, as well as the fact that nearly all of them cause the degradation of materials to a different degree, it can be assumed that these requirements apply to all building elements and built-in materials.

### 1.3. Durability during the Remaining Service Life

During the exploitation of a building object, it is often necessary to determine the so-called remaining period of using the building. It is understood as a period of using a building that exceeds the values found in standards, which is defined by the assumed durability [20,21]. This is needed in order to determine that there is no excessive rapid degradation and that no preventive measures to slow down the degradation rate need to be taken. This finding results from the fact that in the basic period of use, the degradation processes may not be advanced and the symptoms of deterioration may go unnoticed [28]. Usually, in order to assess the advancement of degradation processes, investigations, the scope of which will be different than for severely damaged structures, are needed. When obeying the requirements regarding durability, exploitation properties should not fall below the permissible level, and the structures should only undergo maintenance treatments. The purpose of the technical investigations during normal service life is also different. Within an operation of residential buildings, it is essential to execute a certain assessment of the rate of progressive degradation. This results from the adopted rules of controlling their technical condition, while also taking into account as many factors that affect a building object as possible (their list is included, for example, in ISO 19208 [29]). In buildings designed without taking into account the required durability, and especially in buildings that do not meet durability requirements, i.e., those that were incorrectly designed, constructed or in which the wrong materials were used, unforeseen degradation phenomena may occur, which are difficult to both diagnose and remove. A correct calculation of the remaining service life is mainly possible in the case of properly constructed and designed buildings [30].

British legislation places great emphasis on the correct determination of the remaining service life. This is due to the fact that it was found that large sums of money have been lost in the past as a result of inadequate maintenance and a poorly understood repair scope. This was largely because such a complex problem was dealt with by people without appropriate qualifications. In Great Britain, the so-called “building scientists”, i.e., people with knowledge and experience in the subject of the behavior and changes in the functional properties of apartment houses and their parts under the influence of multiple influences (physical, chemical and biological), play a significant role.

It is important that the assessment of the remaining service life is performed by people with appropriate professional qualifications, especially since in many cases it is necessary to use specialized equipment for non-destructive testing [31]. Such investigations are important because until the real causes of degradation are established, it is unlikely that any preventive action will be effective. In some cases, it can be the cause of damage that requires expensive repairs [32].

The collected data concerning the course of using a residential building is a great help in conducting research. As recommended in BS8210 [33], a property owner should have an archive of all documents related to the building. This is due to the fact that good reports can save building maintenance managers a lot of expense, and can even reduce the risk of the occurrence of errors (caused by the improper diagnosis of the building’s technical condition) during the technical investigations.

It was preliminary noted that the observed technical wear in the initial stage of an element’s exploitation is greater than the theoretical one. After exceeding a certain, determinable time t_p_, the relation is reversed and is valid up to the maximum value of the element’s age t_max_. The difference grows with an increase in the age of the element. This indicates a great imperfection in calculating technical wear using theoretical time methods. The rule is that all the observed building elements represent “overdurability” in the initial stage of their exploitation, and “infradurability” after time t_p_. The aim of the research works presented below is to determine both t_p_ and t_max_ in the context of the durability of building traditional materials.

## 2. Methodology of Research

The preliminary considerations presented the concept of durability with regards to the social aspect of serviceability, which is the most important in the case of the methodology of solutions presented in the article. The technical context of this issue is widely described in the literature (Arendarski’s works [4,5] are fundamental here), and there are huge discrepancies in the recommendations for adopting different values of durability. It seems that for the further analyses carried out in the part of the paper concerning the authors’ own research and model formulation, the presented concept of durability plays a fundamental role with regards to at least three aspects:exploitation, when durability is understood as its exploitation character and when the freedom of its adoption should be specified by assumptions concerning the conditions and level of maintenance of residential buildings;quality, understood as meeting the forecasted needs and requirements of users, and not the expected time when the load-bearing structure of a residential building will reach the ultimate limit state;the remaining service life, in which, after conducting thorough technical examinations of the technical condition of the critical (structurally significant) elements of a residential building, it is possible to make a correction of the durability of the building object’s construction. Such a correction involves the estimation of the period of further use based on the assumptions of the expected level of maintenance, and also the requirements of the users of buildings.

### 2.1. Identification of the Problem

The determination of the degree of the technical wear of the elements of tenement houses is preceded by a decision on the purposefulness and scope of the renovation [34,35]. Such a decision may refer to individual cases, but may also refer to the targeting of actions concerning whole groups of objects. In the latter case, methods that allow for the efficient, and at the same time reliable, estimation of the degree of the technical wear of the building components are needed. There are methods that enable the technical wear of residential buildings, or its elements, to be theoretically determined. However, due to the time of their creation, and the changing conditions of maintaining buildings, it is worth verifying these methods. An attempt to conduct such verification is the subject of this chapter.

Using visual methods, the technical wear was established for 23 selected elements from a homogeneous group of 102 downtown tenement houses, which are characterized by similar architectural, construction and technological solutions—Table 1. Studies carried out at the Institute of Civil Engineering at the Wroclaw University of Science and Technology in 1984–1990 [36] were used in the research. They concerned the appraisal of the technical condition and purposefulness of a major renovation of tenement houses in Wroclaw’s downtown district.

From the above-classified set of 23 elements analysed by a group of experts, a set of 10 elements was selected for further research. These are building elements that meet the validity criterion for the construction of objects and the criterion of the possibility of obtaining reliable test results. The following building elements fulfil the dually defined objective function [37]:Z2—foundations;Z3—basement walls;Z4—solid floors above basements;Z7—structural walls;Z8—inter-story wooden floors (further analysis excludes completely destroyed ceilings on wooden beams, which were exchanged into WPS (beam-and-block floor) or Klein ceilings, as they were heterogeneous elements in the group)Z9—stairs;Z10—roof (rafter framing);Z13—window joinery (further analysis also included windows, which were replaced for new ones of similar construction—of course, verifying the age of the window joinery);Z15—inner plasters;Z20—facades.

The remaining elements were excluded from theoretical and visual investigations due to:Z1—excavations; objections concerning the reliability of measurements (large coefficients of variation);Z5—basement stairs; element of little importance for the structure of an object;Z6—damp-proof insulation; almost the entire sample of buildings does not have both horizontal and vertical damp-proof insulation;Z11—roofing; as a result of the repeated and ad hoc replacement of the roofing or its part, the structure of the homogeneity of the element was disturbed and the age could not be determined;Z12, Z14, Z16, Z17, Z18, Z19—finishing elements—partition walls, door joinery, floorings, painting, kitchens and stoves, metalwork and blacksmith elements; the justification as above, except that it concerns the interference of residents;Z21, Z22, Z23—installations—water–sewage, gas, electricity; an element that has undergone many, and often illegal, arbitrary changes and modifications over the years, etc.—the justification as above.

The result of selecting the 10 “most important” elements of tenement houses, according to the criteria presented above, was to obtain a homogeneous, semantic and numerically consistent database for further studies of the observed states—the results and conclusions of which are presented in the following parts of the article.

### 2.2. States of Technical Wear of Downtown Residential Buildings

#### 2.2.1. Theoretical State

The theoretical methods for the determination of the technical wear Zt of the analysed tenement buildings and their elements involved the linking of it with time parameters, i.e., the period of using a building t, and its expected total durability period T. The durability periods T were adopted according to the study of *Thierry* [6], which presented the experimental determination of the expected service life of old residential buildings—Table 2.

The research included the most frequently used time methods of measuring the technical wear of buildings [5,6,37], which are represented by formulas corresponding to five different classes of the technical wear of building elements:
linear proportionality formula–used in the case of poor maintenance of a building, and corresponds to its bad technical condition (class V, technical wear of 71–100%) and poor technical condition (class IV, technical wear of 51–70%):(1)$$Zt=\frac{t}{T}\;\times 100$$
*Ross or Unger* formula—used in the case of average maintenance of a building, and corresponds to its average technical condition (class III, technical wear of 31–50%):
(2)$$Zt=\frac{t\left(t+T\right)}{2T^{2}}\;\times 100$$
*Romsterfen’s* formula—used in the case of above-average maintenance of a building, and corresponds to its satisfactory technical condition (class II, technical wear of 16–30%):
(3)$$Zt=\frac{t\left(2t+T\right)}{3T^{2}}\;\times 100$$
*Ross and Eytelwein’s* formula—used in the case of very good maintenance of a building, and corresponds to its good technical condition (class I, technical wear of 0–15%):
(4)$$Zt=\frac{t^{2}}{T^{2}}\;\times 100$$

Four time formulas, Equations (1)–(4) were treated as the basic ones. Their application was limited to the determination of the technical wear of the elements of the tested tenement houses with the use of time methods. The remaining six formulas were called supplementary. They include: the *Graff* relationship, which assumes a 70% linear and 30% non-linear course of technical wear; the *Gerard* and the *Hague* formulas, according to which technical wear should not exceed the limit value of 80%; the parametric *Tschellessnigg* function and the “*TEGOVOFA*” formula, which indicate the possibility of an individual course of the wear process by estimating the excess service life of the building. None of the supplementary formulas take into account any additional parameters (apart from t, T) that describe the impact of the maintenance of tenement houses on the size and dynamics of their technical wear. Therefore, it was decided to limit the calculations to the relationships which were described in the most simple way, and most importantly, to those that are assigned to the maintenance states of tenement houses, which were defined in the observed states.

#### 2.2.2. Observed State

The experimental methods of determining the technical wear of downtown tenement houses involved its determination on the basis of actual observations (i.e., an extemporary assessment of a group of experts, measurements with instruments, necessary calculations). It should be remembered that the ultimate goal of the research was to decide on the future of the building, i.e., a technically justified decision on the advisability (technical, economic and social) of modernizing the facility, or conducting preventive renovations, major renovations, or demolitions. The basic premise for such a decision was to estimate the degree of technical wear of the entire tenement house. In order to calculate the technical wear of the building as a whole, a visual method of compensating the degree of wear of individual integrated elements of the building was used [38,39]:(5)$$Ze=\frac{\sum_{i=1}^{n}o_{ei}Z_{ei}}{100}$$
where:Ze—the degree of the technical wear of the building determined using visual methods;o_ei_—the share of the cost of reconstructing the i-th element in the cost of the reconstruction of the entire building [%];Z_ei_—the observed degree of the technical wear of the i-th element [%].

The evaluation result of the technical state of the i-th element by a group of experts was the estimation of the degree of the technical wear within the range from 0 to 100% with a 5% step. In the first stage of developing the research results, the z_ei_ values in the sample of 95–102 buildings were classified into five classes of their technical wear. The classification was conducted using the most popular classification of the technical condition of building elements in the reconstruction process (Table 3), which was proposed by *Winniczek* [37].

The general criteria of the five-step assessment of the technical state of building elements presented in Table 3 were described in more detail with regards to the following types of elements: foundations, structural and partition walls; ceilings; staircases; roofs—construction and roofing; floors and flooring; inner and outer plasters; window and door joinery; stoves and kitchens; installations—water and sewage, gas, central heating and electricity.

The auxiliary criteria prepared in this way were used by a group of experts to assess the degree of the technical wear of 23 elements of downtown tenement houses. In the next stage of developing the research results, which concern each of the 10 elements selected for further analysis, pairs (ti, zi) were sorted in the following three-stage process:first with regards to the increasing values of time ti—different degrees of technical wear were found for the same time values (and vice versa);then with regards to the increasing zi values, by assigning each pair to one of the five states of maintenance of the element;and again with regards to the increasing ti values, however, this time referring to the change in zi in the specific wear class of the element.

The purpose of the above procedure was to prepare the results of the measurements of the degree of technical wear, which was determined according to theoretical formulas and determined using visual methods. For the exemplary structural walls of the aboveground Z7 with a durability of T = 150 years, the youngest and the oldest elements were distinguished out of 102 cases. The differences in the course of the process of their technical wear, which were described using the four theoretical time Equations (1)–(4), as well as the observed state, are shown in Figure 1. This figure contains all the theoretical technical wear data in various technical states of the elements (classes I, II, III, IV-V), and also the observed technical wear that corresponds to the different maintenance conditions of tenement houses (WUI, WUII, WUIII, WUIV, WUV).

### 2.3. Comparative Analysis of Theoretical and Observed States

As to assess the significance of the discrepancies between the values of the theoretical and observed technical wear distribution of elements of downtown residential buildings, two non-parametric tests with the ordinal measurement scale of variables—the *Wilcoxon* test and the *Sign* test—were used for two groups of independent data [40,41]. These tests also involve the proper ordering of ranks, dividing their sums into halves. As a result, the exact probability of the event, in which the obtained statistic from a sample supports the null hypothesis H0 about the identity of the distributions of the variables, was also calculated. If the observed significance level (this exact probability) was lower than the assumed significance level α = 0.05, the null hypothesis was rejected in favour of the alternative hypothesis H1, which showed a significant difference between the distributions.

The *Sign* test is similar to the *Wilcoxon* test [40], and it verifies if the numbers of positive and negative differences between the two variables are identical. The null hypothesis H0 was tested in the same manner as in the *Wilcoxon* test. Although the *Wilcoxon* test is considered to be more powerful, it was decided to include the results of testing using the *Sign* test, which in several cases showed a significant similarity of the studied distributions.

The significance of the differences between the theoretical and observed values of the technical wear distribution of elements of downtown residential buildings was determined using the *Wilcoxon* and *Sign* tests. The observed significance levels, indicating a different (or identical) distribution in all the samples and their subgroups (WU II, WUIII, WUIV), are presented in Table 4.

A comparative analysis of the theoretical and observed technical wear of downtown tenement houses [40,41] was carried out on the basis of a previously prepared database [36]. The comparison of the value of the technical wear of building elements, calculated according to the time formulas—Zt, and determined on the basis of the visual method—Ze, is shown for the aboveground walls Z7 in Figure 2.

The differences between the theoretical technical wear and the wear observed for the age of the elements indicated on the abscissa can be read from the ordinates of the graphs. The comparative analysis was supplemented with the observed technical wear according to the increasing values of the elements’ age (it happens that different values of the technical wear Ze correspond to the same age of the element and the other way round). The prepared values of the function and its arguments (Ze = f{t}) were extended by the theoretical technical wear values calculated according to the time formulas. This enabled the differences DZ = Ze − Zt to be determined as a function of the element’s age.

It should be added that the sample was selected in such a way that the age of three buildings out of 102 buildings significantly deviates from the age ranging from 71 to 130 years, and is equal to 170, 172 and 174 years. As a consequence of such a selection of objects, the oldest of which was erected in 1816, there is a possibility of the occurrence of an apparently paradoxical case when the actual age (technical life) of a residential building exceeds its assumed durability—t_max_ > T. The existence of this fact is used in the next section of the article. However, it must be remembered that the onset of the limiting state of buildings stands for the minimum determined/real durability of their bearing system’s elements like main walls for which t_max_ = T.

The conducted comparative analysis, showing significant differences between the values of the technical wear calculated according to time methods and determined by visual methods, indicated the need to find a way to correct the theoretical formulas. The number of collected observations, which in class III of the technical wear raged from 42 to 83 results for 9 out of 10 selected elements (apart from the wear of the facade Z20), allowed for a statistical examination of the following differences using a normal distribution (n > 30):the difference between the age of the element ti in the i-th moment, and age tp at which the sign of the difference Ze − Zt* changes;the difference between the observed technical wear of element Ze and the theoretical wear Zt* while making a distinction in the meaning and designation of durability:T* = T, as stated by *Thierry* in Table 2T** = t_max_, and then Zt** denotes the technical wear observed for the highest age of an element, which exceeds the durability provided in literature.

Statistical characteristics of the above differences enabled the probabilities of the following random events to be determined as the distribution function Φ [42] of the standardized normal distribution N(m, d)—Table 5:(a)t(i) < t(p) ⇔ P(Ze > Zt*) that is P{t(i) < t(p)} = Φ(Dt < 0) and t(i) > t(p) ⇔ P(Ze < Zt*) that is P{t(i) > t(p)} = Φ(Dt > 0) = 1 − Φ(Dt < 0);(b)DZ* < 0 ⇔ P(Ze < Zt*) = Φ(DZ* < 0) and DZ* > 0 ⇔ P(Ze > Zt*) = Φ(DZ* > 0) = 1 − Φ(DZ* < 0);(c)DZ** < 0 ⇔ P(Ze < Zt**) = Φ(DZ** < 0) and DZ** > 0 ⇔ P(Ze > Zt**) = Φ(DZ** > 0) = 1 − Φ(DZ** < 0).

Calculating the value of the distribution function Φ(Dt) leads to the conclusion (Table 6 and Table 7) that the probability of reaching such an age of the element, in which the observed values of technical wear exceed the theoretical values, is much higher than the probability of the opposite event (except for the basement walls Z3).

The comparative analysis of the distribution function Φ(DZ*) and Φ(DZ**) shows that the probabilities of the theoretical wear values are greater than those observed in the case of the durability of elements that was assumed as in literature (T* = T according to *Thierry*). Moreover, there is an inverse relationship for the durability that corresponds to the age of the oldest element (T** = t_max_).

### 2.4. Analysis of the Residual Term

When estimating the trend model, in addition to finding assessments of structural parameters, some basic parameters of the stochastic structure are also estimated [43,44,45,46,47,48,49]. In turn, the knowledge concerning the evaluation of these parameters allows certain measures of the degree of compliance of empirical data with the data resulting from the trend function to be calculated. This provides the basis for assessing the degree to which accuracy can be assumed in the future on the basis of the estimated trend function.

The basic parameter of the stochastic structure of the trend model that was most correctly selected in the paper was the variance of the random term [d(ξ)]^2^. As the variance of the random term in the general population was not known, it was estimated on the basis of a sample group that had n − k (k = 1) degrees of freedom for the parabolic model. Therefore, it was assumed that the unbiased estimator of the random term is the variance of the residual term, expressed using the following formula:(6)$${\left[d\left(\xi \right)\right]}^{2}=\frac{\sum_{i=1}^{n}{\left(Ze_{i}-\bar{Zt}\right)}^{2}}{n-1}$$

The variance of the residual term is a measure of the order of magnitude of the deviations of the random variables Zei from the trend function.

An additional parameter, supplementing the variance of the residual term, was the mean of deviations of residuals c(ξ) between the measured degree of technical wear of the elements of the analysed buildings and the degree resulting from the model:(7)$$c\left(\xi \right)=\frac{\sum_{i=1}^{n}\left(Ze_{i}-\bar{Zt}\right)}{n}$$

Such a constructed pair of measures of the correctness of selecting the trend function is of significant importance in the analysis of the durability of building elements. When assuming the estimated service life of a tenement house as the independent variable T, the course of the dependence of the variance of residuals and their mean deviations as a function of durability was investigated: [d(ξ)]^2^ = f(T) and c(ξ) = f(T). The variability of these functions for an exemplary element (stairs) is shown in Figure 3. In a correctly constructed model, the variance [d(ξ)]^2^ should reach the minimum for literature values [6,12,13,19,20,21] of durability (T* = 120 years), and the mean of deviations should be close to zero c(ξ). However, depending on the maintenance conditions of stairs, c(ξ) takes zero values and [d(ξ)]^2^ reaches the minimum for T** durability, which is equal to 178 (WUII), 160 (WUIII) and 157 (WUIV) years. The durability of 10 selected elements of downtown residential buildings was therefore analysed with regards to two functions of the trend: for popular parabolic models used to describe theoretical states at work Equations (2)–(4), and for one of the sought models with a significant measure of adjustment (a significant coefficient of determination R^2^).

The variability of these functions for the exemplary element—inter-storey stairs of tenement houses—is presented in Figure 3.

## 3. Results and Conclusions

A repeated regularity is the fact that the observed technical wear in the initial stage of an element’s exploitation is greater than the theoretical one: Ze = f{t} > Zt = f{t}. After overcoming a certain, determinable time t_p_, the relation is reversed and is valid up to the maximum value of the element’s age t_max_. In class III of technical wear, for which the number of qualified building elements is the highest and averages 51, the age from which theoretical technical wear—e.g., structural walls Z7—exceeds the observed wear is equal to 87 years (Figure 2). This difference increases with an increase in the age of the element. This indicates a great imperfection in calculating technical wear using theoretical time methods. The absolute value of the average deviations of the theoretical wear from the observed AVR* (III) is equal to 3.13. The rule is that all the observed building elements show “overdurability” in the initial stage of their exploitation, and “infradurability” after time t_p_. Only in the range determined approximately as (t_p_ − T/10) < t_p_ < (t_p_ + T/10) do the values of the technical wear differ from the theoretical ones by no more than 10%. The greatest number of doubts is raised by the freedom in adopting the durability periods of building elements. The discrepancies are so large that it was decided to use tables developed by *Thierry* and *Zaleski* [6] from 1982, which give values close to the average—e.g., for aboveground structural walls, according to *Thierry,* T* = T and amounts to 130—150 years. After exceeding the value of time t_p_, it seems justified to assume the durability period that corresponds to the highest value of the element’s age for the worst maintenance conditions (WUIV)—which for structural walls is equal to T** = t_max_ =174 years. The result of the attempt to update the assumed estimated durability period is the decrease in the average deviation of the verified values of theoretical technical wear Zt** from the observed technical wear to the value of AVR** (III) = 0.12. The durability of selected elements of the examined buildings, as a posteriori parameter, ranges from 153 to 177 years, and is greater than the corresponding values in literature.

The assessment of the significance of differences between the theoretical and observed arguments of the technical wear distribution of building elements using the *Wilcoxon* and *Sign* tests in most cases confirmed the conclusions of their comparative analysis and showed the significance of differences between the Zt and Ze distributions. This is despite the fact that the *Sign* test indicated that they were identical in the case of the entire distributions (WUI-V) of foundations Z2, basement walls Z3 and construction walls Z7. In turn, both tests (*Wilcoxon* and *Sign* tests) confirmed the identical distribution in individual groups: WUIV—in the case of foundations Z2, basement walls Z3, solid ceilings over the basements Z4, construction walls Z7; WUIII—in the case of construction walls Z7; and WUII—in the case of wooden inter-story ceilings Z8;

In the case of the analysis of the residual term of commonly accepted parabolic models with regards to the assumed durability T of elements of downtown residential buildings, it was noted that:durability T**, which is defined as the variance parameter of residues [d(ξ)]^2^ and which reaches the minimum at a point of the sought durability T**, takes much higher values than literature durability T*;the regularity of the obtained results is surprising due to the fact that the variation range of T** is narrow and ranges from 153 years (for roof structure Z10) to 177 years (for solid ceilings above basements Z4); for comparison, their literature values are assumed as T* = 75 years (Z10) and T* = 150 years (Z4);the reliability of the obtained results of durability T** in the analysis of variance of the residual term was confirmed by studies of exponential and hyperbolic models, in which durability T** (understood as the period of running out of the serviceability of an element (Z = 100%)) assumes values that are similar to those obtained in the analysis of variance of residues (this relationship is only valid for elements with a significant determination coefficient R^2^).

## 4. Concluding Remarks

The fact that there are so many definitions of durability does not entail their proper correlation with the data presented in technical literature [4,5,6,12,37], which are used to determine the durability of building objects. Authors [4,6,7,12,37] usually summarize such data in tables and do not specify the exploitation durability for the data they quote. This is due to the fact that such data often duplicate earlier studies given in domestic and foreign sources, and do not result directly from harmonized European standards concerning the durability of residential buildings erected using traditional methods. 

Issues concerning durability are not adequately considered until the costs associated with the construction and exploitation of buildings have been considered separately. Until recently, the problem of ensuring the durability of buildings was reduced to the issue of protecting against both failures or threats to human life or health. In such a case, efforts were made to provide buildings with maximum and unspecified durability [21,50].

The problems of durability became relevant when the costs related to the construction and maintenance of a facility began to be borne by the investor himself. This also influenced the way of solving durability problems, both during the design stage and also during the exploitation of building objects. Currently, the concept of durability is closely related to the ensuring of appropriate conditions for the use of old tenement houses, as well as to the minimizing of the costs incurred for their maintenance. It is practically impossible to ensure the required durability of residential buildings without providing them with an adequate level of maintenance, which requires the development of a special maintenance and control program. Such a program would have to include the necessary technical inspections of buildings.

One of the basic requirements when determining durability at the stage of designing a building object is the “user’s requirement”. This was provided in ISO 19208 [29], and defined as the needs that must be met by a building. Attention is drawn to the fact that the user can be a person, animal or object, or even technology (in the case of industrial facilities).

Finally, it was proved that in the first period of use of old tenement houses, the phenomenon of “infradurability” occurs, and after exceeding a certain age—depending on the maintenance conditions of the building—the phenomenon of “overdurability” of the building occurs. However, from the point of view of the expected service life of the structure, it is important that the age of a structural element does not approach the assumed durability by less than 25%.

## Figures and Tables

**Figure 1 materials-14-00378-f001:**
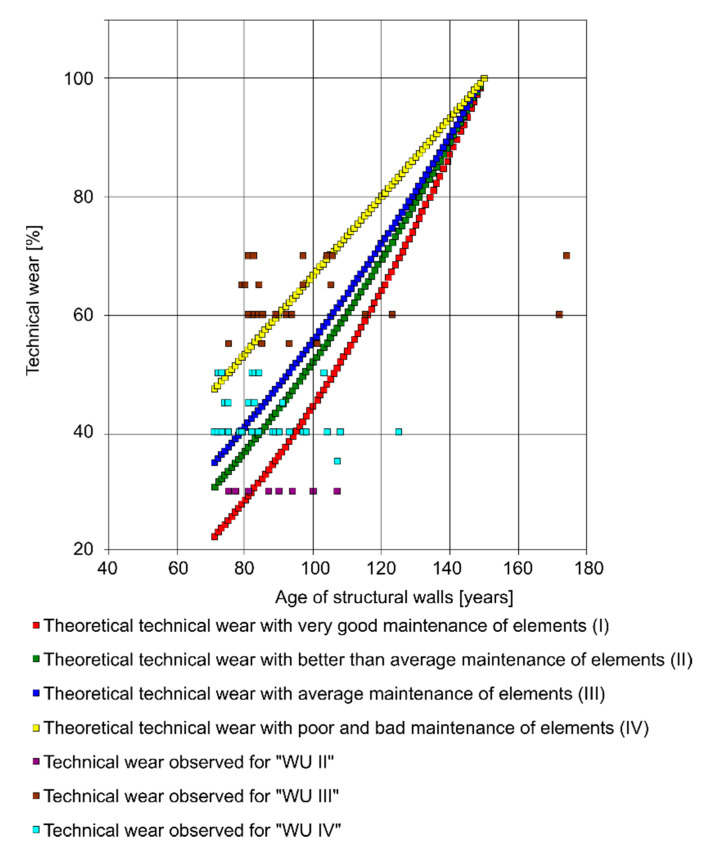
The theoretical and observed technical wear of the aboveground structural walls Z7.

**Figure 2 materials-14-00378-f002:**
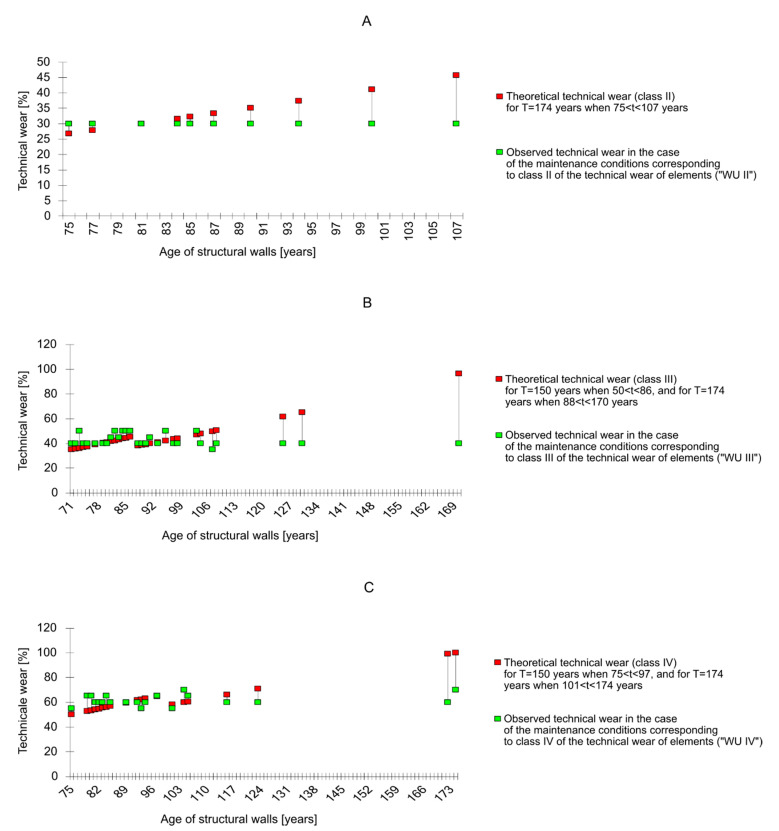
The difference between the theoretical technical wear of the aboveground walls Z7 in the case of: (**A**)—better than average building maintenance; (**B**)—average building maintenance; (**C**)—poor maintenance of the building.

**Figure 3 materials-14-00378-f003:**
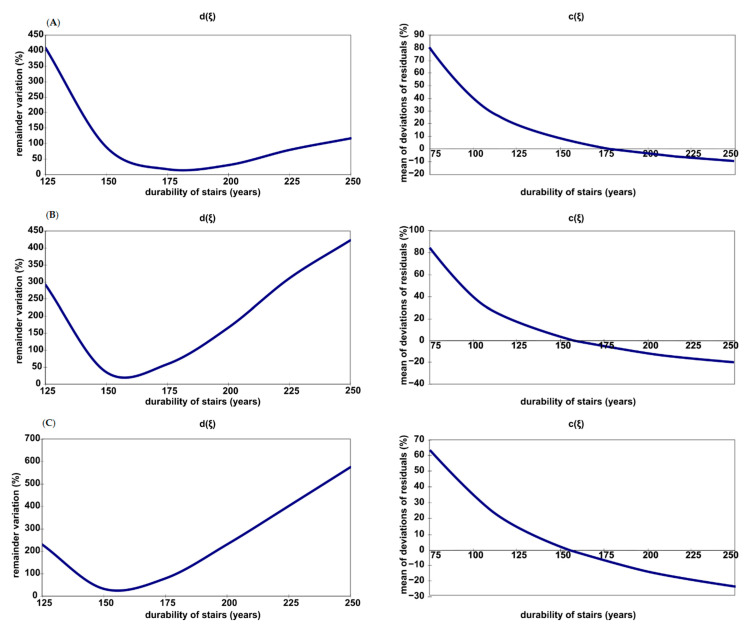
Variance of residuals d(ξ) and mean deviations c(ξ) as a function of durability of inter-storey stairs of tenement houses with: (**A**)—above average maintenance of a tenement house; (**B**)—good maintenance of a tenement house; (**C**)—poor maintenance of a tenement house.

**Table 1 materials-14-00378-t001:** List of elements of downtown tenement houses subjected to a visual inspection of the degree of their technical wear.

Raw State	Building Shell	Finishing State	Installations
Z1 excavations	Z7 structural walls	Z12 partition walls	Z21 water–sewage installations
Z2 foundations	Z8 floors	Z13 window joinery	
Z3 basement walls	Z9 stairs	Z14 door joinery	Z22 gas installations
Z4 floors	Z10 roof/flat roof	Z15 inner plasters	Z23 electrical installations
Z5 stairs	Z11 roofing	Z16 flooring/floors	
Z6 damp-proof insulation	–	Z17 painting	–
	–	Z18 kitchens and stoves	–
–	–	Z19 metalwork— blacksmith elements	–
–	–	Z20 facades	–

**Table 2 materials-14-00378-t002:** Indicative durability periods of selected elements of a residential building, and the building as a whole.

Ceilings		Structural Walls		Stairs	Roof			Installations (Distribution)			Joinery		Plasters		Building
Above Basement	Inter-Story	Basement	Aboveground	Stairs	Construction	Roof Coverings	Flashings	Water and Sewage	Electrical	Gas	Window	Door	Inner	Outer	
a	b	c	d	e	f	g	h	i	j	k	l	m	n	o	p
solid	wooden	brick	brick	steel	wooden	asphalt roll roofing	metal sheet (galvanized)	water and sewage conduits	wires	gas conduits	outer	outer	inner	outer	wooden construction
100–150 years	45–80 years	130–150 years	130–150 years	120–150 years	50–75 years	10–20 years	40–50 years	25–50 years	30–40 years	25–50 years	35–50 years	35–50 years	25–50 years	35–60 years	80–100 years
Klein	WPS	–	–	brick–steel	–	tiles	–	joints	equipment	joints	glazing	inner	–	–	mixed construction
100–130 years	130–150 years	–	–	100–120 years	–	20–60 years	–	20–25 years	25–30 years	20–25 years	20–25 years	40–80 years	–	–	90–120 years
–	Klein	–	–	wooden	–	–	–	equipment	–	–	–	–	–	–	solid construction
–	100–130 years	–	–	20–50 years	–	–	–	15–35 years	–	–	–	–	–	–	100–150 years
–	–	–	–	reinforced concrete	–	–	–	–	–	–	–	–	–	–	–
–	–	–	–	120–150 years	–	–	–	–	–	–	–	–	–	–	–

**Table 3 materials-14-00378-t003:** General criteria for the evaluation and technical classification of the elements of a residential building.

Technical Wear Class of “WU” Element	Condition of Technical Maintenance of the Element	Degree of Technical Wear Resulting from Investigations [%]	Criteria of the Technical Assessment of the Element
I	good	5, 10, 15	Elements of the building structure are well kept and maintained, and do not show wear and damage. The features and properties of the built-in materials meet the standard requirements.
II	satisfactory	20, 25, 30	Building elements are properly maintained. Renovation, which involves minor repairs, additions, maintenance and impregnation, is advisable.
III	average	35, 40, 45, 50	There are minor damages and losses in the building elements, which do not endanger public safety. Partial major renovation is advisable.
IV	poor	55, 60, 65, 70	There are significant damages and losses in building elements. The features and properties of the built-in materials are of a reduced class. Comprehensive renovation, or replacement, is required.
V	bad	75, 80, 85, 90, 95, 100	There are large damages and losses in the building elements that may, or do, pose a risk of the further safe operation of the building. In order to stop the threat, it is necessary to dismantle the element, and construct a new one. In justified cases, the threat may be stopped by the execution of a major and extensive renovation.

**Table 4 materials-14-00378-t004:** The assessment of the significance of differences between the values of distributions Ze and Zt for 10 selected elements of tenement houses.

Appraisal of Differential Significance between Values of Ze and Zt Distributions					
Elements	Test	Class of Technical Wear WU I—V			
		WU I and V	WU II	WU III	WU IV
Z2—foundations	*Wilcoxon test*	R	R	R	I
	*Sign test*	I	R	R	I
Z3—basement walls	*Wilcoxon test*	R	R	R	I
	*Sign test*	I	R	I	I
Z4—solid ceilings above basements	*Wilcoxon test*	R	R	R	I
	*Sign test*	R	R	R	I
Z7—structural walls	*Wilcoxon test*	R	R	I	I
	*Sign test*	I	R	I	I
Z8—inter-storey wooden floors	*Wilcoxon test*	R	I	R	R
	*Sign test*	R	I	R	R
Z9—stairs	*Wilcoxon test*	R	R	R	R
	*Sign test*	R	I	R	R
Z10—roof (rafter framing)	*Wilcoxon test*	R	R	R	R
	*Sign test*	R	I	R	R
Z13—window joinery	*Wilcoxon test*	R	R	R	R
	*Sign test*	R	R	R	R
Z15—inner plasters	*Wilcoxon test*	R	R	R	R
	*Sign test*	R	R	R	R
Z20—facades	*Wilcoxon test*	R	R	R	R
	*Sign test*	R	R	R	R

Explaination: R—distributions are different; I—distributions are identical.

**Table 5 materials-14-00378-t005:** Values of the normal distribution of the theoretical and observed technical wear for 10 selected elements of tenement houses in class III of technical wear.

Class III of Technical Wear (Average Wear)				Distribution Function of the Normal Distribution for the Age of Elements if Ze > Zt* and Ze < Zt*			
No.	Name of Element	Number of Samples	Time of Changing the Sign of the Function	Normal Distribution N(m,d)		Distribution Function F(t)	
				Average Value	Standard Deviation	Probability	Probability
						t(i) < t(p); P(Ze > Zt*)	t(i) > t(p); P(Ze < Zt*)
				(Years)	(Years)	P{t(i) < t(p)} = P(Dt < 0)	P{t(i) > t(p)} = P(Dt > 0) = 1 − P(Dt < 0)
–	–	n	tp	m	d	F(Dt)	1 − F(Dt)
Z2	Foundations	83	86	90.00	17.92	0.911	0.089
Z3	Basement walls	57	93	91.00	20.68	0.038	0.962
Z4	Solid ceilings above basements	45	83	92.00	22.44	0.845	0.155
Z7	Structural walls	61	87	88.00	16.06	0.976	0.024
Z8	Inter-storey wooden floors	54	–	–	–	–	–
Z9	Stairs	49	–	–	–	–	–
Z10	Roof (rafter Framing)	42	47	69.00	30.66	0.737	0.263
Z13	Window joinery	52	30	70.00	20.72	0.527	0.473
Z15	Inner plasters	59	–	–	–	–	–
Z20	Facades	22	–	–	–	–	–

**Table 6 materials-14-00378-t006:** Differences in the distribution function F(DZ*) of theoretical and observed technical wear for 10 selected elements of tenement houses in class III of maintenance.

Class III of Technical Wear (Average Wear)			Distribution Function of the Normal Distribution of Differences: Observed Wear Ze and Theoretical Wear Zt*			
No.	Name of Elements	Number of Samples	Normal Distribution N(m,d)		Distribution Function F(DZ*)	
			Average Value (%)	Standard Deviation (%)	Probability	Probability
					DZ* < 0	DZ* > 0
					P(Ze < Zt*) = P(DZ* < 0)	P(Ze > Zt*) = P(DZ* > 0) = 1 − P(DZ* < 0)
–	–	n	m*	d*	F(DZ*)	1 − F(DZ*)
Z2	Foundations	83	−4.00	13.81	0.892	0.108
Z3	Basement walls	57	−6.00	14.12	0.836	0.164
Z4	Solid ceilings above basements	45	−9.00	16.66	0.794	0.206
Z7	Structural walls	61	−3.00	13.21	0.912	0.088
Z8	Inter-storey wooden floors	54	−54.00	7.23	0.500	0.500
Z9	Stairs	49	−18.00	12.72	0.581	0.419
Z10	Roof (rafter framing)	42	−35.00	39.95	0.692	0.308
Z13	Window joinery	52	−55.00	11.55	0.500	0.500
Z15	Inner plasters	59	−57.00	5.47	0.500	0.500
Z20	Facades	22	−47.00	25.25	0.532	0.468

**Table 7 materials-14-00378-t007:** Differences in the distribution function F(DZ**) of theoretical and observed technical wear for 10 selected elements of tenement houses in class III of maintenance.

Class III of Technical Wear (Average Wear)			Distribution Function of the Normal Distribution of Differences: Observed Wear Ze and Theoretical Wear Zt**			
No.	Name of Elements	Number of Samples	Normal Distribution N(m,d)		Distribution Function F(DZ**)	
			Average Value (%)	Standard Deviation (%)	Probability	Probability
					DZ** < 0	DZ** > 0
					P(Ze < Zt**) = P(DZ** < 0)	P(Ze > Zt**) = P(DZ** > 0) = 1 − P(DZ** < 0)
–	–	n	m**	d**	F(DZ**)	1 − F(DZ**)
Z2	Foundations	83	0.00	10.71	0.000	1.000
Z3	Basement walls	57	−3.00	12.02	0.902	0.098
Z4	Solid ceilings above basements	45	−3.00	14.95	0.920	0.080
Z7	Structural walls	61	0.00	10.39	0.000	1.000
Z8	Inter-storey wooden floors	54	5.00	11.79	0.164	0.836
Z9	Stairs	49	6.00	13.04	0.177	0.823
Z10	Roof (rafter framing)	42	11.00	12.66	0.309	0.691
Z13	Window joinery	52	−54.00	11.50	0.500	0.500
Z15	Inner plasters	59	3.00	15.23	0.079	0.921
Z20	Facades	22	9.00	12.12	0.271	0.729

## Data Availability

No new data were created or analyzed in this study. Data sharing is not applicable to this article.

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
