# Peer review of "Overdurability and Technical Wear of Materials Used in the Construction of Old Buildings"

_materials, 2021, doi:10.3390/ma14020378_

Round 1
Reviewer 1 Report
REVIEW
on the article
Overdurability and Technical Wear of Materials Used in the Construction of Old Buildings
Jarosław Konior
SUMMARY.
The article is devoted to the topical issue of ensuring the technical condition and maintenance of traditional residential buildings that have served for more than 100 years. In many countries around the world, maintaining the proper technical condition of the old housing stock is a problem that affects not only technical but also socio-cultural aspects. Many old houses reflect the cultural traditions of the country and have the status of a cultural heritage site.
The article analyzes the influence of many factors causing their accelerated deterioration of buildings. The estimated durability is a major parameter of the changing age of the apartment buildings under study.
Based on the probabilistic approach, methodological aspects of the technical assessment of the state of the structure have been developed, which should also be aimed at minimizing the subjectivity of expert assessment in the process of technical inspections of residential buildings.
- At the end of the Introduction section, I recommend that the author clearly formulate the purpose of the study.
- It is not clear from the text of the article what is the criterion for the onset of the limiting state of buildings?
- Line 390 - 393. Translate the words czyli and oraz into English.
In general, the article makes a good impression, I recommend it for publication after minimal corrections
Author Response
Dear Reviewer of Materials – MDPI,
Thank you for review of the paper materials-1051216 entitled “Overdurability and Technical Wear of Materials Used in the Construction of Old Buildings” to be published in the Journal “Materials”.
I do appreciate thoughtful and accurate comments presented by the reviewers as well as appreciation of my research works. I have carefully considered a few remarks and have now completed the revisions incorporating the reviewer’s suggestions in the revised uploaded manuscript.
I hope I have taken constructive comments into account in the revised paper and now the final version meets your expectations.
Kind regards,
Jarosław Konior,
Department of Building Engineering, Faculty of Civil Engineering, Wroclaw University of Science and Technology, 50-370 Wrocław, Poland
Here are answers to reviewer’s comments – one by one:
Comment 1: At the end of the Introduction section, I recommend that the author clearly formulate the purpose of the study.
Answer 1: Good point. The purpose of the study has been formulated at the end of Introduction – item 1.3, lines 168 – 175: It was preliminary noted that the observed technical wear in the initial stage of an element's exploitation is greater than the theoretical one. After exceeding a certain, determinable time tp, the relation is reversed and is valid up to the maximum value of the element's age tmax. The difference grows with an increase in the age of the element. This indicates a great imperfection in calculating technical wear using theoretical time methods. The rule is that all the observed building elements represent "overdurability" in the initial stage of their exploitation, and "infradurability" after time tp. The aim of the research works presented below is to determine both tp and tmax in context of durability of building traditional materials.
Comment 2: It is not clear from the text of the article what is the criterion for the onset of the limiting state of buildings?
Answer 2: I assume it is. The onset of the limiting state of buildings stands for the minimum determined / real durability of their bearing system’s elements like main walls for which tmax = T. For better clarity such explanation has been inserted in lines 381 – 383.
Comment 3: Line 390 - 393. Translate the words czyli and oraz into English.
Answer 3: Thanks for spotting. Translated accordingly.
Reviewer 2 Report
General Comments: The MS is a well written and structured report about the technical wear of materials used in residential buildings from Wroclaw. The author compared the course of the theoretical and observed degree of technical degradation of these buildings with their durability. After comprehensive literature review and defining the relationships between durability, maintenance and service life of residential buildings, the author focuses on the technical wear of buildings which was measured by the means of different time methods. In addition, there is final chapter addressing some problems when durability is considered.
In my opinion, the scientific part of the paper is scientifically sound and valuable contribution to the knowledge about the technical wear of materials in the constructions in general.
I do have few minor remarks, most of which are entered in MS-pdf attached.
Structure of the paper:
The structure of the paper is clear and easy to follow. Maybe only at the very end, for me it is a bit ununusal, as the author immediately goes from the methodological work and research to the conclusions. This is followed by a brief summary and discussion. The latter would actually be more accurately described as a concluding remarks. All the needed discussion is actually included in the Ch. 2. and it would be difficult to extract it as a separate chapter.
Order of References:
Normally, works are listed in the text in order. In your case, the works follow in the normal order up to serial number 36, then skip to number 45. Entries follow to parts 37 to 39, then skip again from 39 to 45, then back to parts 40, 41, 42. Again. Skip to part 44, then list part 43 at the very end. Also, you cite 52 papers in the text, but your bibliography only lists 50 works. Where did the two references get lost?
Congratulations on good work, kind regards and good luck!

Author Response
Dear Reviewer of Materials – MDPI,
Thank you for review of the paper materials-1051216 entitled “Overdurability and Technical Wear of Materials Used in the Construction of Old Buildings” to be published in the Journal “Materials”.
I do appreciate thoughtful and accurate comments presented by the reviewers as well as appreciation of my research works. I have carefully considered a few remarks and have now completed the revisions incorporating the reviewer’s suggestions in the revised uploaded manuscript.
I hope I have taken constructive comments into account in the revised paper and now the final version meets your expectations.
Kind regards,
Jarosław Konior,
Department of Building Engineering, Faculty of Civil Engineering, Wroclaw University of Science and Technology, 50-370 Wrocław, Poland
Here are answers to reviewer’s comments – one by one:
Comment 1: Structure of the paper: The structure of the paper is clear and easy to follow. Maybe only at the very end, for me it is a bit unusual, as the author immediately goes from the methodological work and research to the conclusions. This is followed by a brief summary and discussion. The latter would actually be more accurately described as a concluding remarks. All the needed discussion is actually included in the Ch. 2. and it would be difficult to extract it as a separate chapter.
Answer 1: Indeed. Item 3 has been renamed as Results and Conclusions and the latter, final section has been changed into Concluding Remarks
Comment 2: Order of References: Normally, works are listed in the text in order. In your case, the works follow in the normal order up to serial number 36, then skip to number 45. Entries follow to parts 37 to 39, then skip again from 39 to 45, then back to parts 40, 41, 42. Again. Skip to part 44, then list part 43 at the very end. Also, you cite 52 papers in the text, but your bibliography only lists 50 works. Where did the two references get lost?
Answer 2: I fact I am coming back a few times to the previously cited references within text passage, so do not know how the repetitive references may be quoted without skipping in their numbers? Actually, There were cited 51 papers (not 52) in the text; number 51 was a mistake and should have been number 50, therefore has been corrected accordingly.
Reviewer 3 Report
The paper provides a method to estimate the technical wear of different building elements. The technical wear is established for 23 building elements for 102 houses. Finally, the author decided to choose 10 of them for the detailed study. Then, the author compared the degree of the technical wear of the building determined using visual methods with the theoretically determined technical wear. It was concluded that the theoretical wear values are greater than the observation.
The author stated that "Performance characteristics do not relate directly to materials. " which is not understandable and more explanation should be provided. In the paper, the author agreed that the durability of materials determines the durability of the buildings. However, when analyzing the technical wear of each building element, the information about materials was not shown. How does the author consider the factor of materials in the analysis? The paper says "The research included the most frequently used time methods of measuring the technical wear of buildings [5,6,37], which are represented by formulas corresponding to five different classes of the technical wear of building elements ", but actually only four equations are shown here. The reviewer also thinks it is better to provide a figure here to let readers visually understand the differences between them. The paper is well written and well presents. However, the topic does not cover much about materials, so, I believe, it is more suitable for Buildings, but here leaves the decision for the editors.Author Response
Dear Editorial Board of Materials – MDPI,
Thank you for review of the paper materials-1051216 entitled “Overdurability and Technical Wear of Materials Used in the Construction of Old Buildings” to be published in the Journal “Materials”.
I do appreciate thoughtful and accurate comments presented by the reviewers as well as appreciation of my research works. I have carefully considered a few remarks and have now completed the revisions incorporating the reviewer’s suggestions in the revised uploaded manuscript.
I hope I have taken constructive comments into account in the revised paper and now the final version meets your expectations.
Kind regards,
Jarosław Konior,
Department of Building Engineering, Faculty of Civil Engineering, Wroclaw University of Science and Technology, 50-370 Wrocław, Poland
Here are answers to reviewer’s comments – one by one:
Comment 1: The author stated that "Performance characteristics do not relate directly to materials. " which is not understandable and more explanation should be provided.
Answer 1: Indeed, this is misleading and unfortunate statement therefore has been deleted from Introduction as being irrelevant here.
Comment 2: In the paper, the author agreed that the durability of materials determines the durability of the buildings. However, when analyzing the technical wear of each building element, the information about materials was not shown. How does the author consider the factor of materials in the analysis?
Answer 3: Well, it was assumed that considered building elements consist of traditional engineering materials (timber, steel, brick, cement, …) and there is neither point nor topic to describe them technically. The information of buildings materials’ technical state has been presented in the figure 1 of the exemplary element – main walls.
Comment 3: The paper says "The research included the most frequently used time methods of measuring the technical wear of buildings [5,6,37], which are represented by formulas corresponding to five different classes of the technical wear of building elements ", but actually only four equations are shown here. The reviewer also thinks it is better to provide a figure here to let readers visually understand the differences between them.
Answer 3: This is correct as linear proportionality formula is used in the case of a poor maintenance of a building, and corresponds to its bad technical condition (class V, technical wear of 71-100%) and poor technical condition (class IV, technical wear of 51-70%). So, there are four formulas representing classes I, II, III, IV-V of buildings’ technical conditions and the differences between them have been shown in figure 1 (curves yellow, blue, green, and red ones).